# Restricted Random Pruning at Initialization for High Compression Range

**Hikari Otsuka**  *otsuka.hikari@artic.iir.titech.ac.jp*
*Department of Information and Communications Engineering*
*Tokyo Institute of Technology*

**Yasuyuki Okoshi**  *okoshi.yasuyuki@artic.iir.titech.ac.jp*
*Department of Information and Communications Engineering*
*Tokyo Institute of Technology*

**Ángel López García-Arias**  *lopez@artic.iir.titech.ac.jp*
*Department of Information and Communications Engineering*
*Tokyo Institute of Technology*

**Kazushi Kawamura**  *kawamura@artic.iir.titech.ac.jp*
*Department of Information and Communications Engineering*
*Tokyo Institute of Technology*

**Thiem Van Chu**  *thiem@artic.iir.titech.ac.jp*
*Department of Information and Communications Engineering*
*Tokyo Institute of Technology*

**Daichi Fujiki**  *dfujiki@artic.iir.titech.ac.jp*
*Department of Information and Communications Engineering*
*Tokyo Institute of Technology*

**Masato Motomura**  *motomura@artic.iir.titech.ac.jp*
*Department of Information and Communications Engineering*
*Tokyo Institute of Technology*

**Reviewed on OpenReview:** *https://openreview.net/forum?id=yf4ciZcgrg*

## Abstract

Pruning at Initialization (PaI) makes training overparameterized neural networks more efficient by reducing the overall computational cost from training to inference. Recent PaI studies showed that random pruning is more effective than ranking-based pruning, which learns connectivity. However, the effectiveness of each pruning method depends on the existence of skip connections and the compression ratio (the before-after pruning parameter ratio). While random pruning performs better than ranking-based pruning on architectures with skip connections, the superiority without skip connections is reversed in the high compression range. This paper proposes Minimum Connection Assurance (MiCA) that achieves higher accuracy than conventional PaI methods for architectures with and without skip connections, regardless of the compression ratio. MiCA preserves the random connection between the layers and maintains the performance at high compression ratios without the costly connection learning that ranking-based pruning requires. Experiments on image classification using CIFAR-10 and CIFAR-100 and node classification using OGBN-ArXiv show that MiCA enhances the compression ratio and accuracy trade-offs compared to existing PaI methods. In VGG-16 with CIFAR-10, MiCA improves the accuracy of random pruning by 27.0% at $10^{4.7}\times$ compression ratio. Furthermore, experimental analysis reveals that increasing the utilization of the nodes through which information flows from the first layer is essential for maintaining high performance at a high compression ratio.

# 1 Introduction

Although deep neural networks (DNNs) have high generalization capability, both their training and inference are computationally expensive (Arora et al., 2019; Zhang et al., 2019; 2021; Neyshabur et al., 2019; Wen et al., 2022). These high costs arise because their computation depends on a large amount of parameters (Shoeybi et al., 2019; Brown et al., 2020; Dosovitskiy et al., 2021; Woo et al., 2023).

Network pruning achieves high generalization capability despite fewer parameters and can solve this problem. There are various types of pruning, including methods that train while sparsifying the network gradually by penalty terms (Chauvin, 1988; Weigend et al., 1990; Ishikawa, 1996), prune the network after training and then finetune it (LeCun et al., 1989; Hassibi et al., 1993; Lee et al., 2021), and prune and learn iteratively (Frankle & Carbin, 2019; Frankle et al., 2019; Renda et al., 2020). However, these aim to reduce the inference computational cost and need to train the dense model. By contrast, dynamic sparse training (Mocanu et al., 2018; Evci et al., 2020; Jayakumar et al., 2020) and pruning at initialization (PaI) train with sparse networks, thus reducing training computational costs and hence the learning speed can be faster. In particular, PaI has the lowest training computational cost among pruning methods because the network structure is fixed (Price & Tanner, 2021).

Basically, PaI calculates a criterion to determine which parameters are essential and selects the parameters to be pruned based on it. This type of PaI called *ranking-based pruning at initialization* (RbPI) (Lee et al., 2019; Wang et al., 2020; Tanaka et al., 2020) can learn the network connections explicitly but needs to calculate the criterion using an expensive process such as backpropagation. On the other hand, another type of PaI called *random pruning at initialization* (RPI) has a negligibly small additional cost because it only prunes a network randomly without calculating a criterion. At first glance, RbPI seems to perform better than RPI since it learns connections, but some works suggested that RPI could construct subnetworks with similar or better performance obtained by RbPI. Frankle et al. (2021) revealed that RPI and RbPI had comparable accuracy at $1$–$10^2\times$ compression ratios when applying the same sparsity set separately for each layer (i.e., the sparsity distribution). Similarly, the work by Su et al. (2020) showed that RPI with ad-hoc sparsity distribution improved the trade-off between parameter ratio of dense to sparse network— *compression ratio*—and accuracy than RbPI. Furthermore, randomly pruned networks outperform dense networks in aspects such as out-of-distribution detection and adversarial robustness (Liu et al., 2022). Thus, RPI seems to combine simple pruning processing with high performance among PaI methods.

On the other hand, a recent thread of PaI research (Vysogorets & Kempe, 2023) showed a curious phenomenon of RPI: its efficiency at high compression ratios depends on skip connections in the DNN. At more than $10^2\times$ compression ratios, randomly pruned models without skip connections are less accurate than RbPI. It differs from the result in low compression ratio by Frankle et al. (2021) and implicates that highly sparse networks need to learn connection. However, whether connection learning is essential in the high compression range is debatable. For instance, the work by Gadhikar et al. (2023) improved the performance of skip connection-free architecture by adding parameters (i.e., edges) to non-functional neurons in a randomly pruned network at $10$–$10^3\times$ compression ratios. It indicates that even random pruning can improve performance if the connections between layers are preserved. However, this approach is not essential for higher compression ratios because the additional edges inhibit compression. Therefore, there is a need for more essential solutions to improve performance in skip connection-free architecture.

In order to address this problem, this paper introduces a novel PaI algorithm for high compression range: *Minimum Connection Assurance* (MiCA). Specifically, it preserves top-to-bottom information propagation among randomly pruned layers by building a random connection—*minimum connection*—using some of the pre-allocated edges. The minimum connection is constructed by pre-determining and connecting the neurons that the subnetwork uses, and the subnetwork with the connection maintains the pre-defined sparsity distribution even when connecting its neurons randomly. Thus, all allocated edges can be functional even in a high compression range. Since MiCA has this small constraint on the placement of the edges while keeping the connection random, it stands as *restricted* RPI algorithm in the field of PaI algorithms. We evaluate MiCA on image classification using CIFAR-10, CIFAR-100 (Krizhevsky et al., 2009), ImageNet (Russakovsky et al., 2015) and on node classification using OGBN-ArXiv (Hu et al., 2020). In each evaluation, we employ VGG (Simonyan & Zisserman, 2014) and ResNet (He et al., 2016) architectures for the former and graph

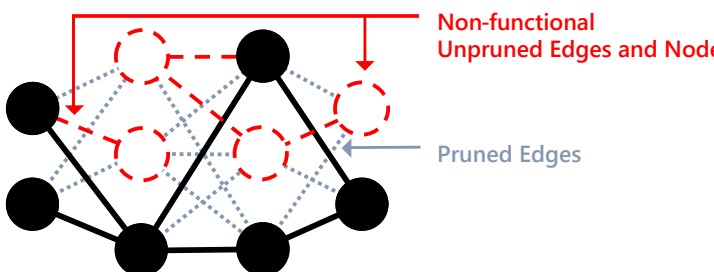

Figure 1: Some edges and nodes stop affecting the output (dashed line) as the network becomes sparser by pruning dotted lines.

convolutional network (GCN) (Kipf & Welling, 2017) and graph isomorphism network (GIN) (Xu et al., 2019) architectures for the latter. MiCA enhances the performance for not only skip connection-free architectures such as VGG but also architectures with skip connections such as ResNet. Furthermore, despite the random connection, MiCA improves the trade-off between compression ratio and inference accuracy compared to RPI and RbPI methods. In other words, MiCA shows that the connections learned by RbPI can be replaced by random connections even in the high-compression range.

The rest of the paper is organized as follows. Section 2 outlines existing PaI methods and describes how to calculate a compression ratio that correctly compares them with MiCA. Then, Section 3 proposes MiCA, and Section 4 compares MiCA with the RPI and RbPI methods. Finally, Section 6 concludes this paper.

## 2 Related Work

PaI algorithms can be categorized into two groups: 1) those that learn the criterion of pruning before training weights (i.e., RbPI); and 2) those that prune randomly (i.e., RPI). The methods included in these groups can be compared by using the compression ratio obtained by eliminating unused edges (Vysogorets & Kempe, 2023). This section recapitulates the method for calculating the corrected compression ratio and outlines the literature on RPI and RbPI.

**Calculation of Corrected Compression Ratios.** Recently, Vysogorets & Kempe (2023) found that PaI algorithms produce significant amounts of redundant parameters that can be removed without affecting the output. Figure 1 illustrates this phenomenon. A pruned network has 10 edges, but the 4 dashed edges do not affect the output. Thus, the apparent compression ratio is $21/10 = 2.1$, but it can also be regarded as $21/(10 - 4) = 3.5$. Correcting the compression ratio calculation by removing them results in a fairer comparison between subnetworks. Subsequent sections use this compression ratio calculation.

**Random Pruning at Initialization (RPI).** RPI prunes each layer randomly based on a pre-defined sparsity distribution calculated by a pre-defined compression ratio. To date, various sparsity distribution design methods have been proposed. For example, Erdős-Rényi-Kernel (ERK) (Evci et al., 2020), which was devised in the context of random graphs, determines the density of $l$-th layer to be proportional to the scale $(C_{in}^{(l)} + C_{out}^{(l)} + k_h^{(l)} + k_w^{(l)})/(C_{in}^{(l)} \times C_{out}^{(l)} \times k_h^{(l)} \times k_w^{(l)})$, where $C_{in}^{(l)}$, $C_{out}^{(l)}$, $k_h^{(l)}$, and $k_w^{(l)}$ denote input channels, output channels, kernel height, and kernel width of the $l$-th layer, respectively. Ideal Gas Quotas (IGQ) (Vysogorets & Kempe, 2023) focuses on the fact that traditional global pruning methods (Lee et al., 2019; 2021; Tanaka et al., 2020) intensively remove parameter-heavy layers. It determines the constant $F$ based on the target compression ratio and calculates the density of $l$-th layer as $\left(Fe^{(l)} + 1\right)^{-1}$, where $e^{(l)}$ is the number of edges in the $l$-th layer. The subnetworks to be pruned based on these sparsity distributions achieve comparable or better performance against RbPI and RPI using other distributions (Vysogorets & Kempe, 2023). However, those with skip connection-free architectures cannot achieve such performance as the compression ratio increases. Although some solutions, such as adding edges and resampling (Gadhikar et al., 2023), are proposed to address this RPI weakness, they are impossible at higher compression ratios or inefficient. Unlike these methods, our approach is efficient and works at higher compression ratios.

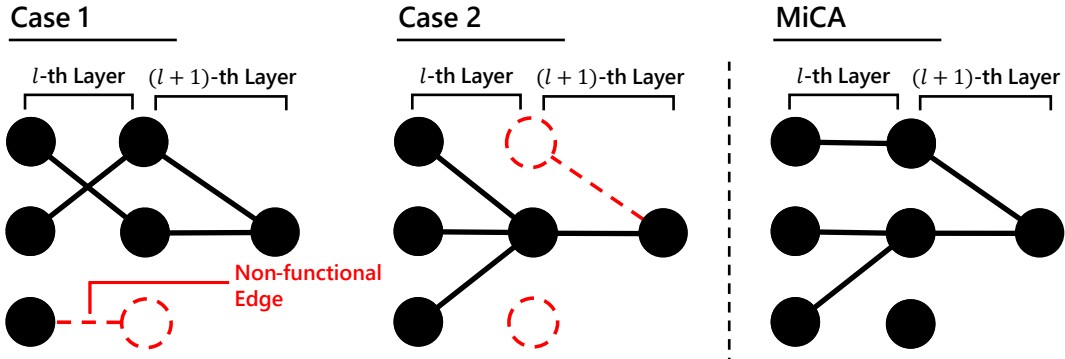

Figure 2: In cases 1 and 2, randomly pruning the network can make some edges non-functional. On the other hand, MiCA keeps all edges functional while allowing randomness in the connections.

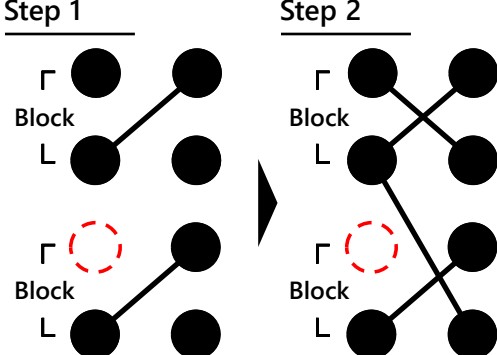

Figure 3: Edge placement procedures to construct a minimum connection. First, an input node of each block is connected to an output node, and then the remaining unconnected nodes are connected.

It constructs a subnetwork with only pre-allocated edges, except that pre-defined sparsity distributions invariably cause non-functional neurons (i.e., nodes). Moreover, our approach does not require the iterative pruning operation.

**Ranking-Based Pruning at Initialization (RbPI).** RbPI determines the pruning priorities based on the initial state of a network and a dataset. For example, SNIP (Lee et al., 2019) uses the magnitude of the backpropagation gradient after one iteration as a parameter's pruning priority. GraSP (Wang et al., 2020) prunes the edges that do not reduce the gradient flow of a subnetwork for a dataset preferentially. SynFlow (Tanaka et al., 2020) updates a parameter's pruning priority by using the $l_1$-path norm of a network (Neyshabur et al., 2015) as a loss and prunes iteratively without dataset. Unlike other RbPI methods, it can avoid layer-collapse (Hayou et al., 2020) at high compression ratios. As seen from these, RbPI takes into account the information flow of the initialized network for the pruning criteria. Hence, its sparse sub-networks tend to connect layers to each other. On the other hand, RbPI requires pre-training to calculate priorities, which is prohibitively expensive. RPI and our approach have a lower cost because they do not need to calculate priorities.

## 3 MiCA: Minimum Connection Assurance

As demonstrated by Gadhikar et al. (2023), maintaining connections in each layer can improve the sparse network's accuracy after training, especially when the compression ratio is high. This section proposes MiCA, a method to make all pre-allocated edges as functional as possible with a few operations based on the fact that the connections at each layer are guaranteed if all pre-allocated edges are functional. MiCA

---

**Algorithm 1** Number of Output Nodes Analysis

---

**Require:** Pre-defined maximum number of output nodes $n_{out}^{(L)}$, number of edges $e^{(1)}, e^{(2)}, ..., e^{(L)}$, maximum block size $b^{(1)}, b^{(2)}, ..., b^{(L)}$.

1: **procedure** NODEANALYSIS($n_{out}^{(L)}, e^{(1)}, e^{(2)}, ..., e^{(L)}, b^{(1)}, b^{(2)}, ..., b^{(L)}$)

2:      $n_{out,ideal}^{(L)} \leftarrow \min\left(n_{out}^{(L)}, e^{(L)}\right)$

3:      **for** $l = L$ **to** 2 **do**

4:         **if** $b^{(l)} \geq n_{out,ideal}^{(l)}$ **then**

5:            $n_{out,ideal}^{(l-1)} \leftarrow x \in \left\{ x \in \mathbb{N} \;\middle|\; \left(e^{(l)}/n_{out,ideal}^{(l)}\right) \leq x \leq \left(e^{(l)}/b^{(l)}\right) \right\}$

6:         **else**

7:            $n_{out,ideal}^{(l-1)} \leftarrow \left\lceil e^{(l)}/n_{out,ideal}^{(l)} \right\rceil$

8:         **end if**

9:      **end for**

10:      **return** $n_{out,ideal}^{(1)}, n_{out,ideal}^{(2)}, ..., n_{out,ideal}^{(L)}$

11: **end procedure**

---

first constructs a random connection named minimum connection from the input to the output layer using some pre-allocated edges. Then, it builds a subnetwork by randomly connecting the nodes involved in the connection. The presence of minimum connection ensures that randomly placed edges become functional. In order to construct the minimum connection, it is necessary to determine how many nodes should be used at each layer to ensure that all placed edges are functional. Therefore, we first introduce an analytical method for determining the ideal number of nodes and then describe how to construct the minimum connection.

**The ideal number of nodes analysis:** A layer has non-functional edges if and only if the number of nodes used in the adjacent layers is too large or too small relative to the number of edges pre-assigned to that layer. Figure 2 exemplifies the two situations: in case 1, the edges of $l$-th layer connect to all nodes of $(l+1)$-th layer. It causes some non-functional edges at $l$-th layer if the number of edges of the $(l+1)$-th layer is low; in case 2, all edges connect to a few nodes of the next layer. It causes some non-functional edges at $(l+1)$-th layer if the number of edges of the $(l+1)$-th layer is high. Thus, based on a pre-defined sparsity distribution and each layer architecture, we pre-determine which nodes to use to avoid these situations.

We consider pruning a neural network constructed with $L$ layers. We first define the maximum number of input and output nodes pre-defined as a network structure of $l$-th layer as $n_{in}^{(l)}$ and $n_{out}^{(l)}$, respectively. For instance, in a convolutional layer with input channels $C_{in}^{(l)}$, output channels $C_{out}^{(l)}$, kernel height $k_h^{(l)}$, and kernel width $k_w^{(l)}$, $n_{in}^{(l)}$ and $n_{out}^{(l)}$ are $C_{in}^{(l)} \times k_h^{(l)} \times k_w^{(l)}$ and $C_{out}^{(l)}$, respectively. Similarly, in a fully-connected layer with input features $f_{in}^{(l)}$ and output features $f_{out}^{(l)}$, $n_{in}^{(l)}$ and $n_{out}^{(l)}$ are $f_{in}^{(l)}$ and $f_{out}^{(l)}$, respectively. Furthermore, we denote the number of input and output nodes of the pruned $l$-th layer as $V_{in}^{(l)} \subseteq \{1, 2, ..., n_{in}^{(l)}\}$ and $V_{out}^{(l)} \subseteq \{1, 2, ..., n_{out}^{(l)}\}$, respectively. Additionally, we define a *block* $V_{blk,i} \subseteq V_{in}^{(l)}$ in the $l$-th layer as the input nodes corresponding to an output node in the $(l-1)$-th layer. There are $\left|V_{out}^{(l-1)}\right|$ blocks in $l$-th layer, and each size of block ranges from 1 to $b^{(l)}$, where $b^{(l)}$ is defined as follows:

$$b^{(l)} := \begin{cases} n_{in}^{(l)}/n_{out}^{(l-1)} & (l > 1) \\ k_h^{(1)} \times k_w^{(1)} & (l = 1 \text{ and first layer is a convolutional layer}) \\ 1 & (\text{otherwise}) \end{cases} \tag{1}$$

For example, if both the $(l-1)$-th and $l$-th layers are convolutional layers, the maximum size $b^{(l)}$ is $k_h^{(l)} \times k_h^{(l)}$, and it is 1 if the $(l-1)$-th layer is a fully-connected layer. As a particular case, we also define a block with a maximum size $b^{(1)}$ in the first layer. If the first layer is a convolutional layer, we consider a kernel as a block; otherwise, we consider a single neuron as a block.

The pruned $l$-th layer can be regarded as a bipartite graph connecting nodes in $V_{in}^{(l)}$ and $V_{out}^{(l)}$. To ensure that all edges of $l$-th layer and adjacent layers become functional, the number of edges $e^{(l)}$ in the $l$-th layer

---

**Algorithm 2** Step 1 of Minimum Connection Construction at $l$-th layer

---

**Require:** Network $N$, input nodes $V_{in}^{(l)}$, pre-defined number of edges $e^{(l)}$, the ideal number of output nodes $n_{out,ideal}^{(l)}$ pre-calculated by Algorithm 1.

1: **procedure** STEP1($N, V_{in}^{(l)}, e^{(l)}, n_{out,ideal}^{(l)}$)
2:      $V_{used} \leftarrow \{\}$
3:      $i_{out} \leftarrow 1$
4:      $V_{out}^{(l)} \leftarrow \{\}$
5:      **for** $V_{blk,i}^{(l)}$ **in** blocks of $l$-th layer **do**
6:          $v_{in}^{(l)} \leftarrow x \in V_{blk,i}^{(l)}$
7:          $V_{used} \leftarrow V_{used} \cup \left\{ v_{in}^{(l)} \right\}$
8:          **if** $i_{out} \leq n_{out,ideal}^{(l)}$ **then**
9:              $v_{out}^{(l)} \leftarrow i_{out}$
10:            $i_{out} \leftarrow i_{out} + 1$
11:          **else**
12:              $v_{out}^{(l)} \leftarrow x \in \left\{ x \in \mathbb{N} \mid 1 \leq x \leq n_{out,ideal}^{(l)} \right\}$
13:          **end if**
14:          Connect $v_{in}^{(l)}$ and $v_{out}^{(l)}$ in $N$
15:          $V_{out}^{(l)} \leftarrow V_{out}^{(l)} \cup \left\{ v_{out}^{(l)} \right\}$
16:          $e^{(l)} \leftarrow e^{(l)} - 1$
17:          **if** $e^{(l)} \leq 0$ **then**
18:              **break**
19:          **end if**
20:      **end for**
21:      **return** $N, e^{(l)}, V_{used}, i_{out}, V_{out}^{(l)}$
22: **end procedure**

---

must satisfy $\max\left( \left| V_{in}^{(l)} \right|, \left| V_{out}^{(l)} \right| \right) \leq e^{(l)} \leq \left| V_{in}^{(l)} \right| \left| V_{out}^{(l)} \right|$. Solving this inequality for $\left| V_{in}^{(l)} \right|$ leads to

$$\frac{e^{(l)}}{\left| V_{out}^{(l)} \right|} \leq \left| V_{in}^{(l)} \right| \leq e^{(l)}. \tag{2}$$

Additionally, considering that up to $b^{(l)}$ input nodes of the $l$-th layer connect to an output node of the $(l-1)$-th layer for $1 < l \leq L$, $\left| V_{in}^{(l)} \right|$ satisfies the inequality

$$\left| V_{out}^{(l-1)} \right| \leq \left| V_{in}^{(l)} \right| \leq \left| V_{out}^{(l-1)} \right| \times b^{(l)}. \tag{3}$$

From these two inequalities, we need to satisfy

$$\frac{e^{(l)}}{\left| V_{out}^{(l)} \right|} \leq \left| V_{out}^{(l-1)} \right| \leq \frac{e^{(l)}}{b^{(l)}} \tag{4}$$

for $1 < l \leq L$ so that all edges in the network become functional.

We sequentially analyze the ideal number of output nodes $n_{out,ideal}^{(l)}$ from the output layer to the input layer. We select $n_{out,ideal}^{(l)}$ randomly from the range $\frac{e^{(l)}}{n_{out,ideal}^{(l)}} \leq n_{out,ideal}^{(l-1)} \leq \frac{e^{(l)}}{b^{(l)}}$ based on Equation 4. Note that we define $n_{out,ideal}^{(L)} := \min\left( n_{out}^{(L)}, e^{(L)} \right)$. If $n_{out,ideal}^{(l)} < b^{(l)}$, we set $n_{out,ideal}^{(l-1)}$ as $\left\lceil \frac{e^{(l)}}{n_{out,ideal}^{(l)}} \right\rceil$ to minimize the loss of input nodes. In cases where the network features a branching structure, such as residual connections,

---

**Algorithm 3** Step 2 of Minimum Connection Construction at $l$-th layer

---

**Require:** Network $N$, input nodes $V_{in}^{(l)}$, pre-defined number of edges $e^{(l)}$, the number of output nodes $n_{out,ideal}^{(l)}$ pre-calculated by Algorithm 1.

1: **procedure** $\text{STEP2}(N, V_{in}^{(l)}, e^{(l)}, n_{out,ideal}^{(l)})$
2:     $N, e^{(l)}, V_{used}, i_{out}, V_{out}^{(l)} \leftarrow \text{STEP1}(N, V_{in}^{(l)}, e^{(l)}, n_{out,ideal}^{(l)})$
3:     **while** $e^{(l)} > 0$ **and** $\left(|V_{used}| < \left|V_{in}^{(l)}\right| \text{ or } i_{out} \leq n_{out,ideal}^{(l)}\right)$ **do**
4:         **if** $|V_{used}| < \left|V_{in}^{(l)}\right|$ **then**
5:             $v_{in}^{(l)} \leftarrow x \in V_{in}^{(l)} \setminus V_{used}$
6:             $V_{used} \leftarrow V_{used} \cup \left\{v_{in}^{(l)}\right\}$
7:         **else**
8:             $v_{in}^{(l)} \leftarrow x \in V_{in}^{(l)}$
9:         **end if**
10:        **if** $i_{out} \leq n_{out,ideal}^{(l)}$ **then**
11:            $v_{out}^{(l)} \leftarrow i_{out}$
12:            $i_{out} \leftarrow i_{out} + 1$
13:        **else**
14:            $v_{out}^{(l)} \leftarrow x \in \left\{x \in \mathbb{N} \mid 1 \leq x \leq n_{out,ideal}^{(l)}\right\}$
15:        **end if**
16:        Connect $v_{in}^{(l)}$ and $v_{out}^{(l)}$ in $N$
17:        $V_{out}^{(l)} \leftarrow V_{out}^{(l)} \cup \left\{v_{out}^{(l)}\right\}$
18:        $e^{(l)} \leftarrow e^{(l)} - 1$
19:    **end while**
20:    **return** $N, e^{(l)}, V_{out}^{(l)}$
21: **end procedure**

---

we choose the larger $n_{ideal}^{(l-1)}$ obtained at each branch to maintain connections across all input nodes. The computational cost of this analysis, as depicted in Algorithm 1, is negligibly small as it relies solely on simple operations with pre-defined constants.

**Minimum connection construction:** Upon determining the ideal number of output nodes used in each layer, the minimum connection construction is finalized through edge placement so that it is satisfied that $\left|V_{out}^{(l)}\right| = n_{ideal}^{(l)}$. We establish a minimum connection in two steps, outlined in Figure 3:

1. Select an input node from a block $V_{blk,i}^{(l)}$ and connect it to an unconnected output node. This process is iterated for all blocks (for the detailed procedure, see Algorithm 2).

2. Connect the remaining unconnected nodes. If the minimum degree of input or output nodes is already 1, we connect the unconnected nodes randomly to other nodes (for the detailed procedure, see Algorithm 3).

These steps are executed sequentially from the input to the output layer. Finally, MiCA places the remaining edges randomly within the nodes of the minimum connection, as depicted in Algorithm 4. These edges are functional regardless of placement, owing to the minimum connection.

## 4 Experiments and Results

This section evaluates MiCA using the experimental setup described in Section 4.2. It shows that MiCA performs better than conventional RPI and RbPI methods, especially in the high compression range and regardless of whether skip connections exist.

---

**Algorithm 4** Minimum Connection Assurance

---

**Require:** Network $N$, the maximum number of input nodes $n_{in}^{(1)}$, the maximum number of output nodes $n_{out}^{(L)}$, pre-defined number of edges $e^{(1)}, e^{(2)}, ..., e^{(L)}$, maximum block size $b^{(1)}, b^{(2)}, ..., b^{(L)}$.

1: $n_{out,ideal}^{(1)}, n_{out,ideal}^{(2)}, ..., n_{out,ideal}^{(L)} \leftarrow \text{NODEANALYSIS}(n_{out}^{(L)}, e^{(1)}, e^{(2)}, ..., e^{(L)}, b^{(1)}, b^{(2)}, ..., b^{(L)})$

2: $V_{in}^{(1)} \leftarrow \left\{ 1, 2, ..., n_{in}^{(1)} \right\}$

3: **for** $l = 1$ **to** $L$ **do**

4:      $N, e^{(l)}, V_{used}, i_{out}, V_{out}^{(l)} \leftarrow \text{STEP2}(N, V_{in}^{(l)}, e^{(l)}, n_{out,ideal}^{(l)})$

5:      **while** $e^{(l)} > 0$ **do**

6:         $v_{in}^{(l)} \leftarrow x \in V_{in}^{(l)}$

7:         $v_{out}^{(l)} \leftarrow x \in V_{out}^{(l)}$

8:         Connect $v_{in}^{(l)}$ and $v_{out}^{(l)}$ in $N$

9:         $e^{(l)} \leftarrow e^{(l)} - 1$

10:      **end while**

11:      **if** $l < L$ **then**

12:         $V_{in}^{(l+1)} \leftarrow (l+1)$-th layer input nodes connected to $V_{out}^{(l)}$

13:      **end if**

14: **end for**

---

### 4.1 Notation of Methods

We briefly introduce the notation of the methods compared in the subsequent experiments in advance. RPI uses a pre-defined sparsity distribution for ERK, IGQ, SNIP, GraSP, and SynFlow. Here, we consider the sparsity distribution of the network pruned by SNIP, GraSP, and SynFlow, which are RbP, as a pre-defined sparsity distribution. RPI methods with these distributions are denoted **RPI-ERK**, **RPI-IGQ**, **RPI-SNIP**, **RPI-GraSP**, and **RPI-SynFlow**, respectively. Similarly, MiCA using these distributions are denoted **MiCA-ERK**, **MiCA-IGQ**, **MiCA-SNIP**, **MiCA-GraSP**, and **MiCA-SynFlow**, respectively. For the RbPI experiments, SNIP, GraSP, and SynFlow are chosen as RbPI methods, and these are specified as **RbPI-SNIP**, **RbPI-GraSP**, **RbPI-SynFlow**, respectively. For more detail on each RPI and RbPI method, see Section 2.

### 4.2 Experimental Settings

This paper evaluates MiCA on two classification tasks: image classification and node classification. The latter is one of the major tasks for graph neural networks (GNN), and this paper covers it to investigate the versatility of our approach for general tasks that do not use convolutional architectures.

We employ the CIFAR-10, CIFAR-100, and ImageNet datasets in image classification. For CIFAR-10 and CIFAR-100, 40,000 images are used as training data and 10,000 as validation data, while we use the default set split for ImageNet. The architectures used in the image classification experiments are VGG-16, ResNet-20, and ResNet-50, and each implementation is based on the code provided by Tanaka et al. (2020). In particular, VGG-16 includes a batch normalization layer and removes the bias of the convolutional layer. All experiments for this task use stochastic gradient descent (SGD) applying Nesterov's acceleration method (Nesterov, 1983) with a momentum of 0.9. CIFAR-10 and CIFAR-100 experiments are run five times with a batch size 128 for 160 epochs, and the ImageNet experiment is run once with a batch size 256 for 90 epochs. For VGG-16, the weight decay is set to 0.0001, and the learning rate is started at 0.1 and multiplied by 0.1 after 60 and 120 epochs. For ResNet-20, the weight decay is set to 0.0005, and the learning rate is started at 0.01 and multiplied by 0.2 after 60 and 120 epochs. For ResNet-50, the weight decay is set to 0.0001, and the learning rate is started at 0.1 and multiplied by 0.1 after 30, 60, and 80 epochs.

On the other hand, in node classification, we experiment with the OGBN-ArXiv dataset split by default proportion. We employ the GCN and GIN architectures, which are variants of GNN, with four layers and implement each architecture based on the codes provided by Wang et al. (2019); Huang et al. (2022). We

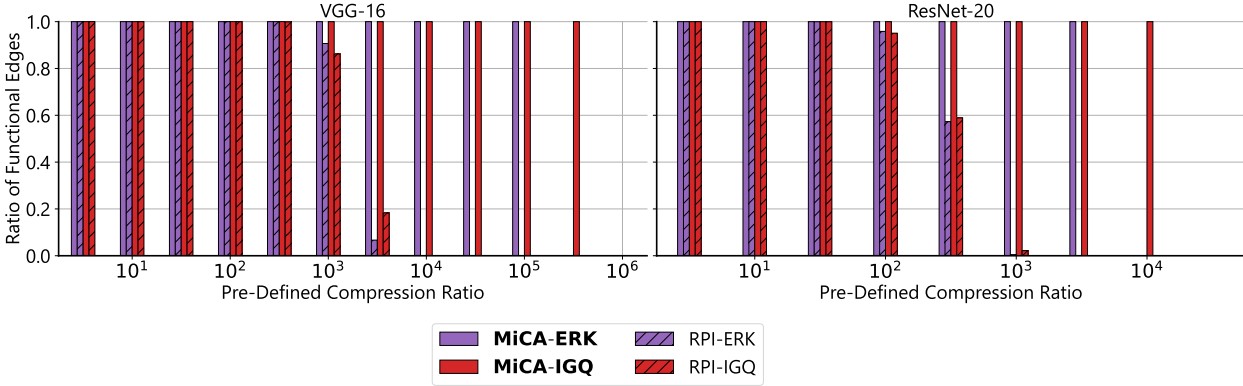

Figure 4: Comparison of the ratio of functional edges between RPI and MiCA with ERK and IGQ for pre-defined compression ratios. Unlike other RPI methods, MICA makes all edges functional, even at a high compression ratio.

use Adam (Kingma & Ba, 2014) and cosine learning rate decay for the node classification. Each architecture is set to an initial learning rate of 0.001 and is trained for 400 epochs. Also, these experiments are run five times.

For all experiments, we use values $10^{0.5}, 10^1, ..., 10^{5.5}$, and $10^6$ as pre-defined compression ratios. Note that results are not plotted if all edges are non-functional after compression. SNIP and GraSP use $10\times$ the training data amount relative to the dataset's number of classes and a batch size of 128. SynFlow prunes the initialized network for 100 iterations.

## 4.3 Pre-Defined Sparsity Distribution Maintenance of MiCA

This section shows how much MiCA and RPI leave functional edges for each pre-defined compression ratio. The result is plotted in Figure 4. The ratios of functional edges for RPI-ERK and RPI-IGQ begin to decrease around $10^3\times$ compression ratio for VGG-16 and $10^2\times$ for ResNet-20. On the other hand, MiCA continues to use all the pre-allocated edges as much as possible and maintains the sparsity distribution even in the high compression range. MiCA-ERK and MiCA-IGQ have almost all functional edges even at $10^5\times$ compression ratio for VGG-16 and $10^{3.5}\times$ compression ratio for ResNet-20. However, all edges can be non-functional at high compression ratios, as seen in MiCA-ERK for VGG-16 at $10^{5.5}\times$ compression ratio. This phenomenon is caused by the way the pre-defined sparsity distribution is designed. Some pre-defined sparsity distributions allocate no edges to a few layers when the pre-defined compression ratio is exceptionally high. As a result, all edges are non-functional, regardless of how they are placed.

## 4.4 MiCA vs. Random Pruning

Figure 5 compares MiCA and RPI with ERK, IGQ, SNIP, GraSP, and SynFlow. First, we focus on the ERK and IGQ experiments of the two left columns. In the VGG-16 experiment (first column), RPI-ERK and RPI-IGQ have a sharp performance drop for $\geq 10^3\times$ compression ratios for both CIFAR-10 and CIFAR-100. These with a pre-defined compression ratio of $10^{3.5}\times$ have an actual compression ratio of $> 10^4\times$, and those with a pre-defined compression ratio of $10^4\times, 10^{4.5}\times, ..., 10^5\times$, and $10^6\times$ cannot be plotted as all edges are non-functional (i.e., the corrected compression ratio is infinite). On the other hand, MiCA-ERK and MiCA-IGQ maintain the same compression ratio as the pre-defined compression ratio and suffer less performance degradation. Note that MiCA-ERK is not plotted for $10^{5.5}\times$ and $10^6\times$ compression ratios. This phenomenon is due to ERK's design, as mentioned in Section 4.3, and the same occurs in other sparsity distributions (e.g., IGQ) and other experiments. As shown in Figure 5 (a), MiCA-IGQ achieves an accuracy of 44.4% for $10^{4.5}\times$ compression ratio, significantly higher than RPI-IGQ's accuracy of 12.6% for $10^{4.3}\times$ compression ratio. This result suggests that the minimum connection supports learning in the high compression range.

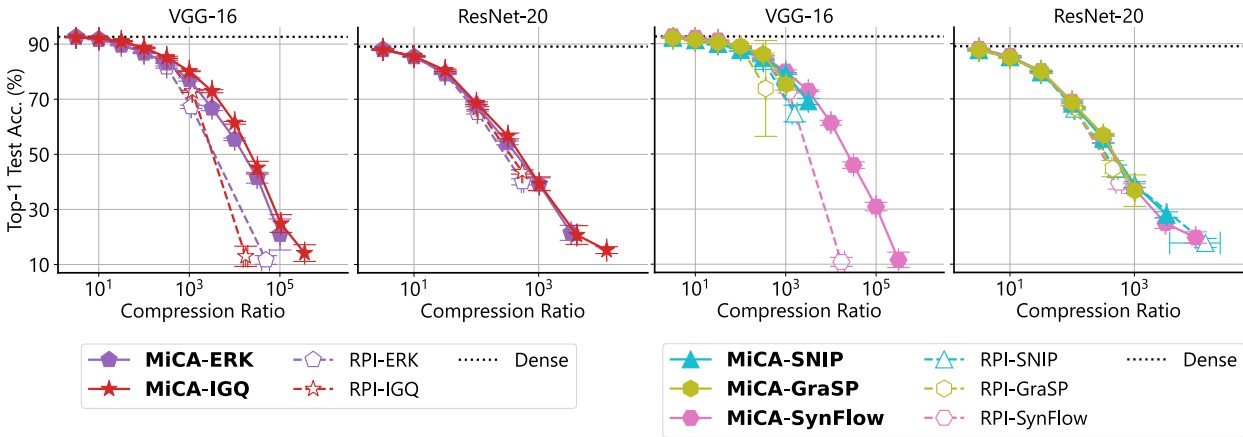

(a) CIFAR-10 experiments.

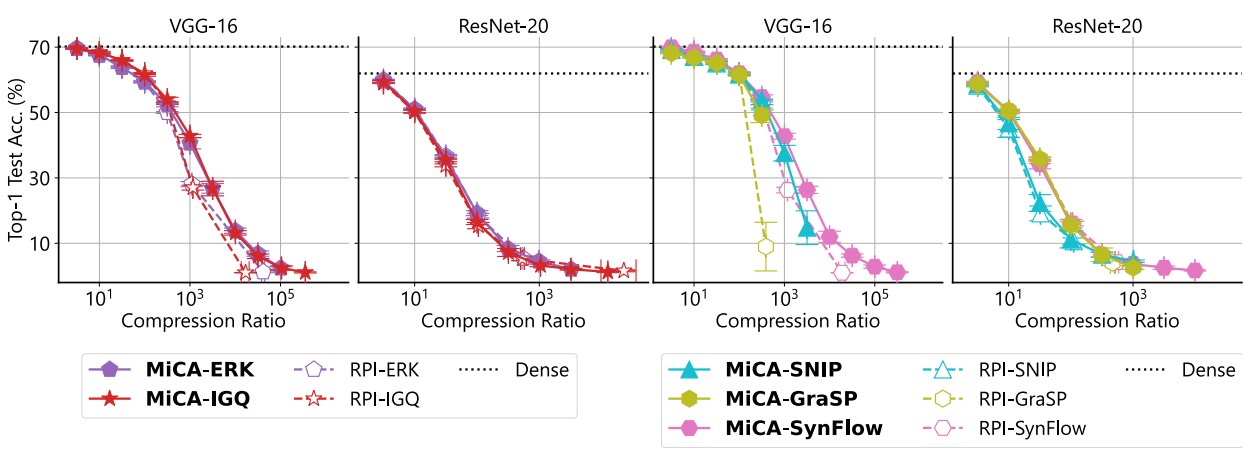

(b) CIFAR-100 experiments.

Figure 5: Comparison of accuracy between RPI and MiCA on CIFAR-10 and CIFAR-100. In VGG-16, MiCA reduces performance degradation in the high compression range and improves the accuracy and compression ratio trade-off. It also shows slight performance improvement for the sparsity distributions such as IGQ and SynFlow in ResNet-20 experiments using CIFAR-10. Note that some plots are not plotted because the corrected compression ratio is infinite.

In the ResNet-20 experiment (second column), the performance difference between MiCA and RPI is less drastic than in the VGG-16 experiment. In particular, CIFAR-100 experiments (Figure 5 (b)) show little difference. However, it is hardly surprising considering that skip connections help randomly pruned networks learn in the high compression range (Hoang et al., 2023). For CIFAR-10 experiments (Figure 5 (a)), MiCA-ERK slightly improves the trade-off between compression ratio and accuracy against RPI-ERK in the compression range of $10^2$–$10^3\times$. ResNet-20 has several layers that do not have skip connections, and it is therefore considered that the minimum connection supports learning in those layers.

Then, we state the results of the right two columns in Figure 5. In the VGG-16 experiment (third column), MiCA-SNIP, MiCA-GraSP, and MiCA-SynFlow significantly improve the trade-off between accuracy and compression ratio as in the ERK and IGQ experiments. It is particularly evident in the CIFAR-100 experiment (Figure 5 (b)). Thus, MICA overcomes the performance degradation at a high compression range for skip connection-free architectures, regardless of the pre-defined sparsity distribution. However, the performance improvements in the ResNet-20 (fourth column) are still minute. This result seems regrettable but highlights the importance of skip connections to RPI.

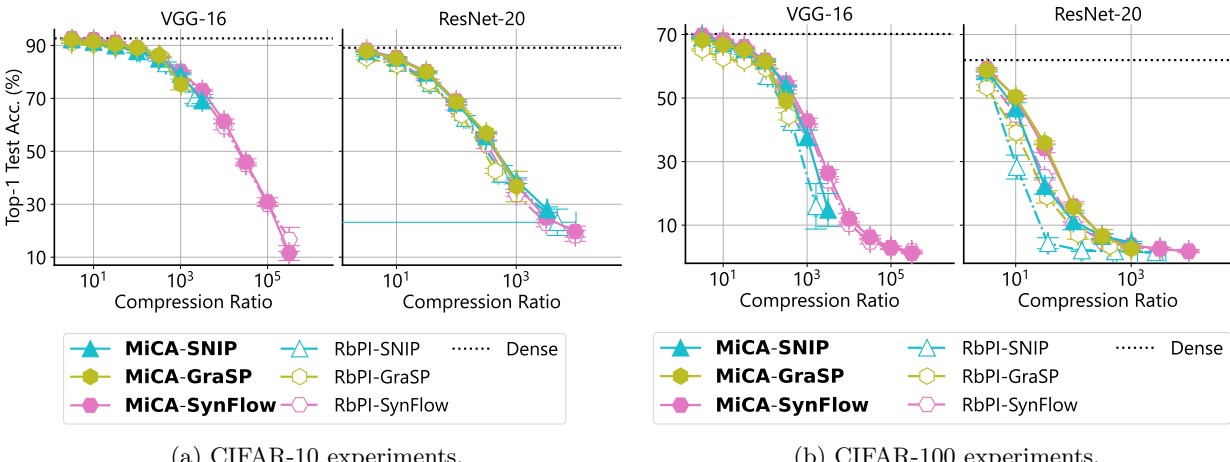

(a) CIFAR-10 experiments.

(b) CIFAR-100 experiments.

Figure 6: Comparison of accuracy between RbPI and MiCA. MiCA improves the accuracy and compression ratio trade-off more than RbPI in ResNet-20. MiCA also performs as well as RbPI in the VGG-16 experiments. Interestingly, MiCA outperforms RbPI in both VGG-16 and ResNet-20 experiments using CIFAR-100.

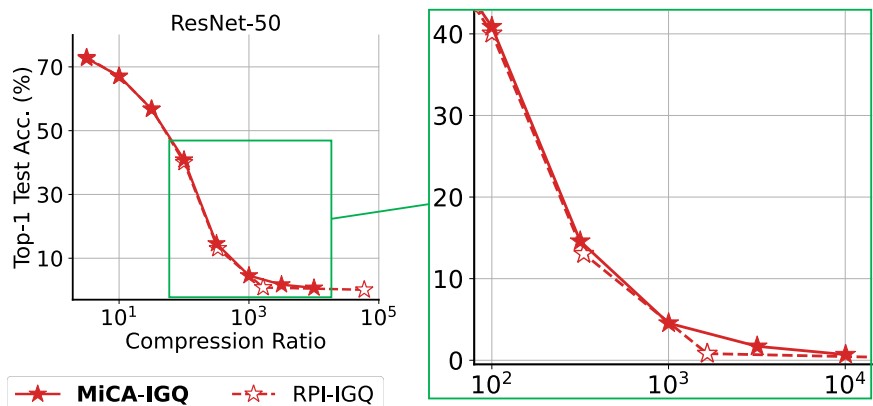

Figure 7: Comparison of accuracy between RPI-IGQ and MiCA-IGQ using ResNet-50 on ImageNet. MiCA improves the accuracy and compression ratio trade-off slightly.

## 4.5 MiCA vs. Ranking-Based Pruning

Figure 6 compares the compression ratio and accuracy between RbPI and MiCA on CIFAR-10 (left two columns) and CIFAR-100 (right two columns). We employ SNIP, GraSP, and SynFlow as RbPI methods. Despite random connections, MiCA achieves comparable performance to RbPI in VGG-16. Although it has already been observed that randomly pruned networks achieve performance comparable to RbPI (Frankle et al., 2021), to the best of our knowledge, this is the first time that the same result is reported in the high compression range. In other words, this result shows that RbPI only learns layer-to-layer connections, not high-performance subnetworks, regardless of the compression ratio. Interestingly, the ResNet-20 experiments show a more pronounced performance difference than the RPI result in Figure 5. It suggests that MiCA's RPI aspect helps it to maintain higher performance than RbPI in ResNet-20 because MiCA connects each layer randomly. Also, RbPI may have significant variations in compression ratio in the high compression range. On the other hand, MiCA is robust regarding compression ratio and can achieve the pre-defined compression ratio.

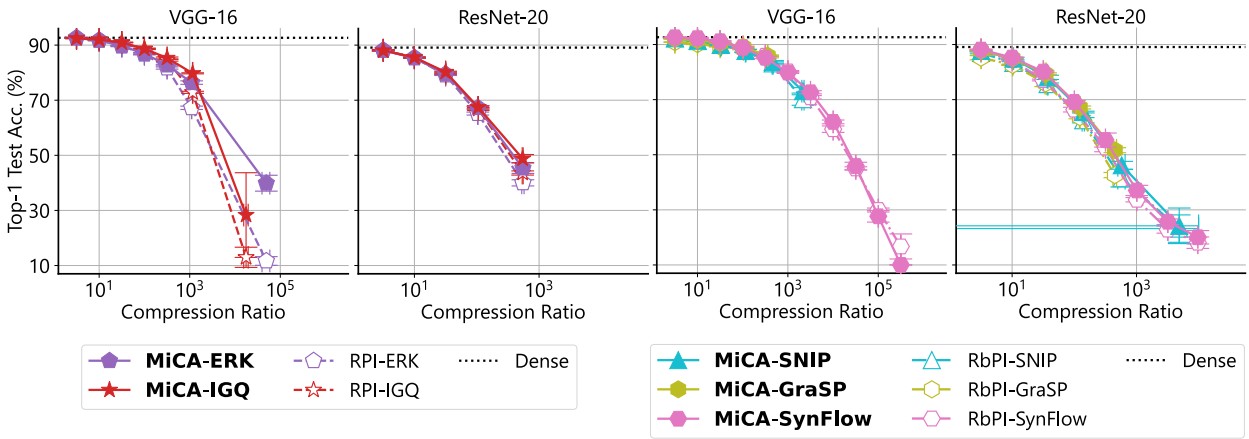

(a) CIFAR-10 experiments.

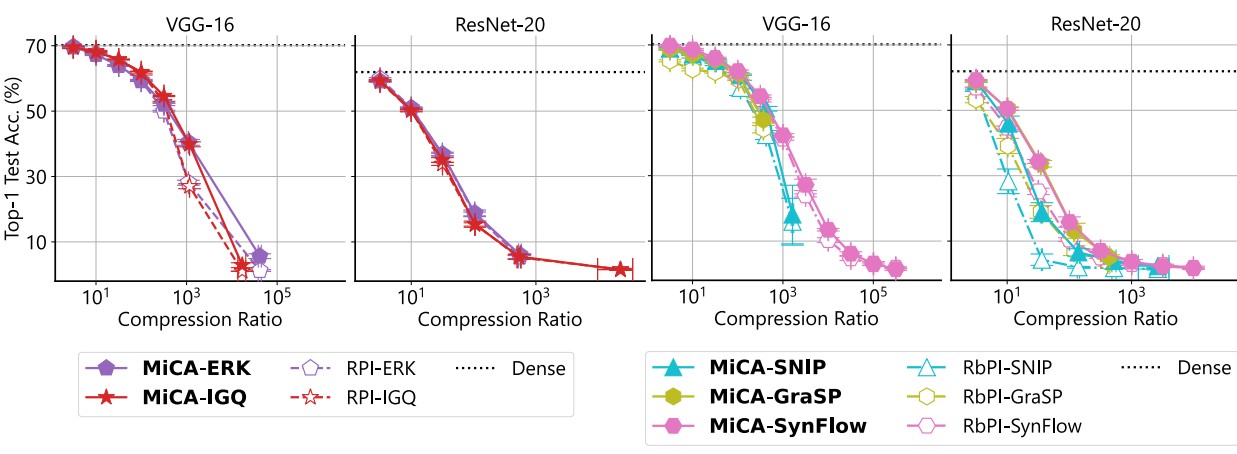

(b) CIFAR-100 experiments.

Figure 8: Comparison of RPI and MiCA with the same corrected sparsity distribution on CIFAR-10 and CIFAR-100. When the number of parameters used in each layer is matched, MiCA performs better than other PaI methods when the compression ratio is high.

## 4.6 ImageNet Experiments

Figure 7 compares the compression ratio and accuracy between RPI-IGQ and MiCA-IGQ on ImageNet in the high compression range. MiCA-IGQ improves the accuracy and compression ratio trade-off slightly over RPI-IGQ. As in the CIFAR-10/100 experiments, the accuracy improves even in the architecture with skip connections, suggesting the importance of the connection relationships between layers. RPI-IGQ has the same accuracy as a random network at a pre-defined compression ratio of $10^4 \times$. However, MiCA-IGQ maintains higher accuracy than a random network even at a pre-defined compression ratio of $10^{4.5} \times$. Thus, maintaining the layer-wise connections enables one to learn from the data, even in the case of an extremely high compression ratio.

## 4.7 Performance Comparison for the Same Sparsity Distribution

Previous sections match the pre-defined sparsity distribution of MICA and other PaI methods and compare each method. In contrast, this section evaluates the corrected sparsity distribution of RPI and RbPI as the pre-defined sparsity distribution of MiCA. Then, we show that the network structure constructed by MiCA

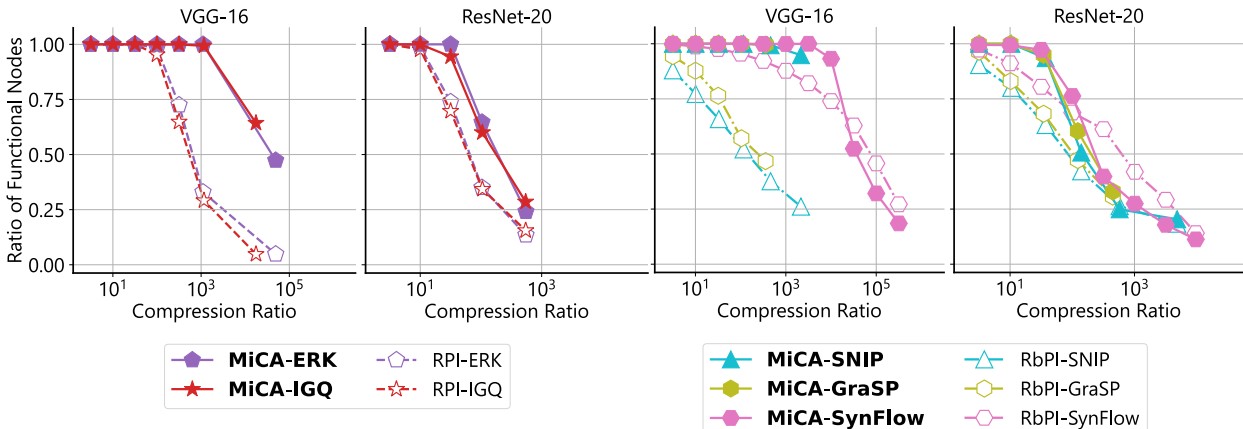

Figure 9: Comparison of the compression ratio with the ratio of functional nodes among the nodes flowing information from the first layer. MiCA maintains a high functional node ratio even in a high compression range compared to other PaI methods. Exceptions are RbPI-SynFlow, which retains a higher ratio than MiCA at high compression ratios.

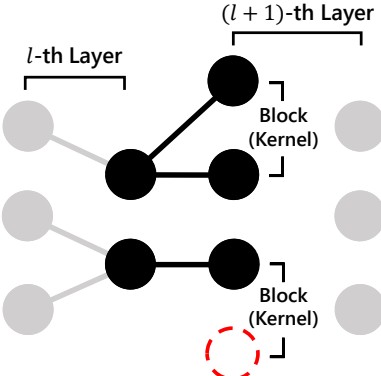

Figure 10: In the $l$-th convolutional layer, each output node has connections with up to $b^{(l+1)} = k_h^{(l+1)} \times k_w^{(l+1)}$ input nodes in the $(l+1)$-th convolutional layer. These connections, formed during the convolution process, may persist even after pruning, making input nodes of (l+1)-th layer non-functional.

achieves higher performance against other PaI methods, even when the number of edges at each layer is matched.

Figure 8 compares RPI, RbPI, and MiCA for the same sparsity distribution. The two columns on the left show RPI experiments, while the two on the right show RbPI experiments. Even when using the same sparsity distribution, MiCA improves the accuracy more than RPI (left two columns). The results in VGG-16 on CIFAR-10 are remarkable: MiCA-ERK achieves 27.0% higher accuracy than RPI-ERK with $10^{4.7}\times$ compression ratio. Performance improvements of MiCA can also be seen in ResNet-20 at a high compression ratio. Both RPI and MiCA ought to make similar networks due to the random connection in this situation, but the performance difference is more noticeable when the compression ratio is high. In addition, the subnetworks built by RbPI suffer more performance degradation than those built by MiCA, regardless of compression ratio and architecture (right two columns). However, in the VGG-16 experiments (third column), RbPI-SynFlow is more accurate than MiCA in the compression range above $10^5\times$. At $10^{5.5}\times$ compression ratio in CIFAR-10 experiments, MiCA-SynFlow achieves 10.0%, which is not different from the random performance, while RbPI-SynFlow achieves 20.3%. Furthermore, these accuracies are almost identical for CIFAR-100 experiments. In other words, RbPI-SynFlow is superior to MiCA only when the compression ratio is extremely high and the architecture does not have skip connections.

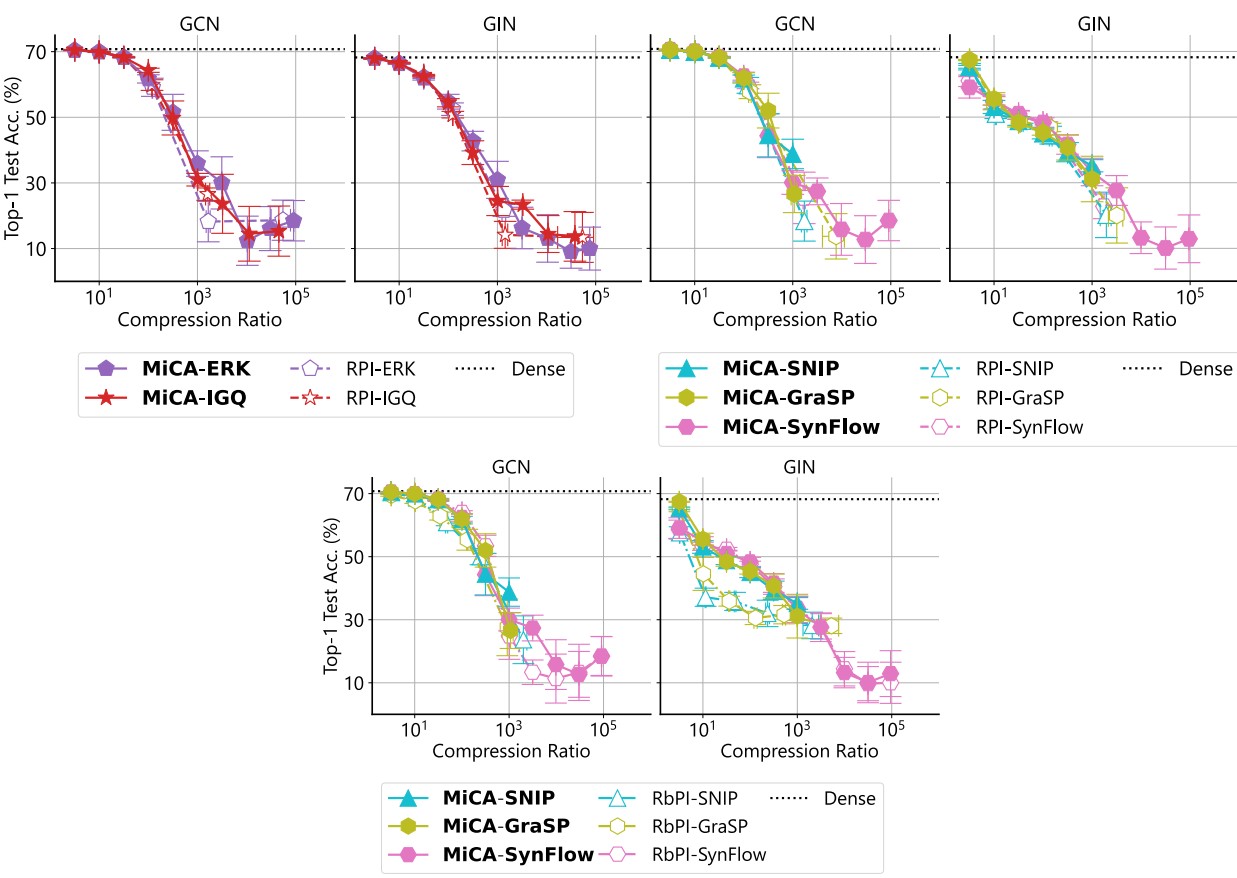

Figure 11: Comparison of the accuracy between RPI, RbPI, and MiCA on OGBN-ArXiv with same pre-defined sparsity distribution.

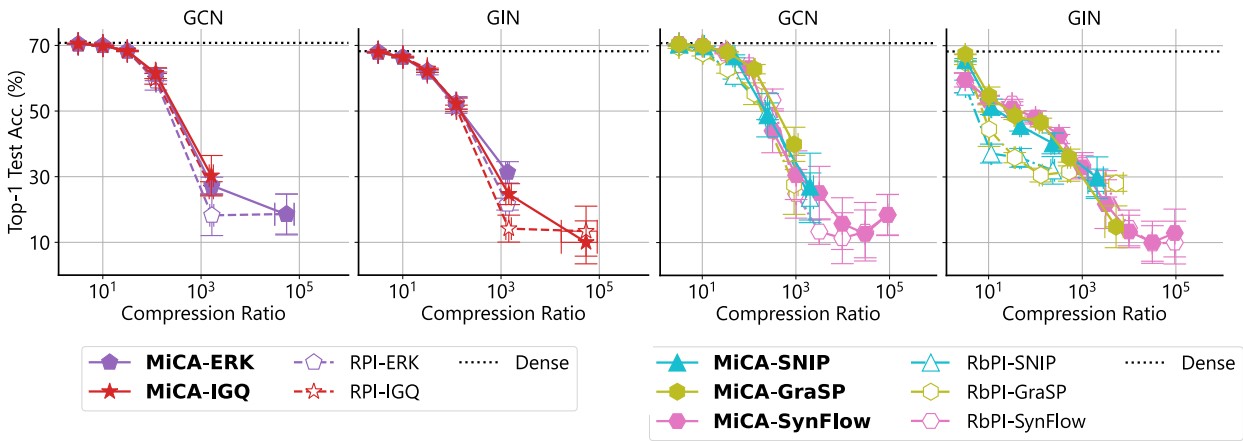

Figure 12: Comparison of RPI, RbPI, and MiCA accuracy on OGBN-ArXiv with the same corrected sparsity distribution.

## 4.8 Top-to-Bottom Information Propagation

This section shows that MiCA propagates information without loss compared to other PaI methods. As in the previous section, the pre-defined sparsity distribution of MICA is matched to the corrected sparsity distribution of each method.

Figure 9 compares the ratio of functional nodes to all nodes that information flows from the first layer for each compression ratio. While RPI and RbPI methods reduce the ratio of functional nodes as the compression ratio increases, MiCA maintains the high ratio of functional nodes. In particular, RPI-ERK and RPI-IGQ have almost 3/4 of the nodes non-functional in VGG-16 at $10^3\times$ compression ratio, whereas MiCA-ERK and MiCA-IGQ keep all nodes functional. Interestingly, RbPI-SynFlow keeps the ratio of functional nodes relatively higher than other PaI methods without MiCA, even in the high compression range. In contrast, RbPI methods make nodes non-functional even in the low compression range. At the compression ratio of $> 10^4\times$ in VGG-16 and $> 10^2\times$ in ResNet-20, RbPI has a higher ratio of functional nodes than MiCA. Given that RbPI-SynFlow achieves comparable or higher accuracy than MiCA at $\geq 10^5\times$ compression ratio in VGG-16 (Figure 8) and that RbPI builds a specific network at the skip connection-free architecture (Hoang et al., 2023), it suggests that sparse networks with a high ratio of functional nodes maintain accuracy at a high compression ratio. MiCA makes nodes non-functional in a high compression range, but the ratio is lower than in other methods. At $10^{4.7}\times$ compression ratio in VGG-16, MiCA-ERK keeps nearly 50% of the nodes functional, whereas RPI-ERK keeps most of the nodes non-functional. In the ResNet-20 experiments, MiCA also has a higher ratio of functional nodes than other methods, but the difference is lower than in VGG-16. It shows that information from the first layer flows to the subsequent layers even after pruning due to skip connections. In other words, the top-to-bottom information flow is narrowed by pruning, but skip connections allow it to flow to the subsequent layers.

Why do some nodes become non-functional even after recalculating the compression ratio? This is because the convolution process produces non-functional nodes. The convolution process connects one input node in $l$-th convolutional layer with several output nodes in $(l+1)$-th convolutional layer, as shown in Figure 10. This connection does not use network parameters; hence, it is preserved after recalculating the compression ratio. Consequently, nodes can be non-functional if the number of remaining edges in $(l+1)$-th layer is small.

## 4.9 Experiments with Non-Convolutional Networks on Node Classification

Through the preceding experiments, we have observed that MiCA is effective in convolutional structures. However, what about for other architectures? This section examines the effectiveness of MiCA using architectures for other tasks that do not involve convolutional networks. Specifically, we experiment with node classification using GCN and GIN, both of which are MLP-based architectures. Note that while GCN seems to have the convolutional layers from the name, the network structure that processes the features is an MLP.

Figure 11 and 12 compare the accuracy between RPI, RbPI, and MiCA on OGBN-ArXiv with the same predefined and corrected sparsity distribution, respectively. Similar to the convolutional network experiments, MiCA maintains high accuracy, comparable to RPI, in the low compression range and enables training similar to RbPI in the high compression range. In other words, MiCA demonstrates the effects of restricted RPI, which lies between RPI and RbPI, even in non-convolutional networks.

# 5 Limitations and Implications of This Work

This section discusses the limitations of this work and the implications of the proposed MiCA.

**Limitations:** Our work has several limitations: 1) As shown in the ImageNet experiment with ResNet-50, the impact of applying MiCA in the context of large-scale datasets is minimal. We have also conducted experiments with other settings using ImageNet, but results beyond the ResNet-50 experiment have not yet been verified. Thus, there is potential for further enhancement in handling large-scale datasets. 2) While the average accuracy improves, some variation in accuracy must be tolerated due to the random selection of non-pruned edges. Further analysis and refinement are necessary to minimize the accuracy variance by tightening the restrictions in the pruning procedure. 3) Since our work focuses on high compression ranges, accuracy is inevitably lower than in low compression cases. Thus, we must explore whether MiCA has advantages in the trade-off between accuracy and computational complexity, particularly considering sparse matrix operations.

**Implications:** Our proposed method MiCA has several implications: 1) Our work offers new insights into the underlying mechanisms governing the functionality of a network pruned by the PaI method under ultra-sparse conditions. The field of PaI is still insufficiently researched under such circumstances, and our work implies an investigation into uncharted territory within the field. 2) The accuracy improvement achieved with MiCA in the high compression range highlights the significance of focusing on layer-wise connectivity for enhanced performance in ultra-sparse networks. Our work demonstrates that accuracy enhancement can be attained solely through simple node connectivity without relying on data or other costly information (e.g., gradients). Although this paper has not demonstrated practical-level accuracy in the high compression range, this empirical finding by MiCA within the field of PaI holds significance, particularly considering the recent trend of over-parameterization and the potential importance of sparse network architectures in the future.

## 6 Conclusion and Future Work

This paper introduces Minimum Connection Assurance, a novel approach to PaI methods. It addresses the critical issue of achieving higher compression ratios while preserving model accuracy, particularly in the high compression range. MiCA enhances the accuracy and compression ratio trade-off, surpassing the performance limitations encountered by RPI on skip connection-free architectures.

Our experiments not only validate the efficacy of MiCA in achieving high compression ratios but also shed light on the underlying factors that govern accuracy after training, which are present within the network created by PaI. Notably, MiCA demonstrates the ability of random connections to substitute for those learned by RbPI, especially in the high compression range. That is a phenomenon previously observed only in the low compression range. In other words, this paper first reveals that learned connections are less than or equivalent to random connections at any compression ratio. Furthermore, our analysis suggests that it is necessary to make all nodes from which information flows functional to improve accuracy.

Our finding has potential for future applications in the area of large-scale models such as foundation models (Bommasani et al., 2021) and large language models (Brown et al., 2020). As the size of state-of-the-art models grows, the associated training and inference costs become increasingly prohibitive. In order to reduce these costs, model compression needs to be sufficiently higher before training than the present. In this respect, our finding in the high compression range can be expected to be important in the future. Moreover, it also has the potential contribution in areas such as fine-tuning large models to downstream tasks by pruning (Jiang et al., 2022).

A promising avenue for future work lies in exploring the appropriate setting method of pre-defined sparsity distributions depending on the minimum connection. While MiCA demonstrates superior performance to conventional PaI methods, the accuracy is contingent upon the pre-defined sparsity distribution. Investigating novel sparsity distributions based on the minimum connection while maintaining critical connections could unveil further optimizations in model compression.

## Acknowledgement

This work was supported in part by JSPS KAKENHI Grant Number JP23H05489.

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
