# OpenReview forum: "Restricted Random Pruning at Initialization for High Compression Range"
_TMLR — Accepted by TMLR_

### Review · Reviewer_vomQ · 2024-02-03

**Summary Of Contributions:**

The paper introduces Minimum Connection Assurance (MiCA), a novel pruning at initialization (PaI) method for neural networks, aimed at maintaining high performance at high compression ratios without the need for costly connection learning. MiCA preserves random connections between layers while ensuring minimum necessary connections are maintained, enabling better accuracy and efficiency, particularly in architectures with and without skip connections across various compression ratios. Experimental results on CIFAR-10 and CIFAR-100 datasets demonstrate MiCA's superiority over conventional PaI methods, improving compression ratio and accuracy trade-offs significantly. The study suggests a future direction in exploring sparsity distributions tailored for MiCA to further enhance performance.

**Audience:**

Yes

**Broader Impact Concerns:**

1. Although being effective, the proposed subnetwork construction (two steps on page 5) is still ad hoc, which may not be principled enough to be adopted in different settings (models, datasets).
2. The novelty and impact will be limited if the method can only work for convolutional layers.

**Claims And Evidence:**

Yes

**Requested Changes:**

1. The proposed subnetwork construction method is only studied for convolution layers. I don't see why this method cannot be extended to transformers.
2. This method "sequentially analyze the number of connectable nodes  from  the output layer to the input layer." This analysis will not be scalable once we switch to large-scale models of billions of parameters or hundreds of layers.
3. This work only test CIFAR-10 and CIFAR-100, which is limited.

**Strengths And Weaknesses:**

1. The motivation of this paper is well described.
2. Authors provided extensive experiments.

---

> ### Author Response · Authors · 2024-02-17
> **Official Comment by Authors**
>
> Thank you for your positive feedback and constructive and helpful comments. We would like to respond to your comments as follows:
>
> > The proposed subnetwork construction method is only studied for convolution layers. I don't see why this method cannot be extended to transformers.
>
> Thank you for pointing this out to us. Our method is not only applicable to convolutional layers, but also to general architectures (including transformers). We have revised the manuscript to make this point clear.
>
> > This method "sequentially analyze the number of connectable nodes from the output layer to the input layer." This analysis will not be scalable once we switch to large-scale models of billions of parameters or hundreds of layers.
>
> We are sorry that this part was not clear in the original manuscript. To make it clear that the analysis is only a simple calculation that can be applied to the large-scale models, we have added algorithmic steps to the manuscript (Algorithm 1).
>
> > This work only test CIFAR-10 and CIFAR-100, which is limited.
>
> Thank you for pointing this out. We have added the results of ImageNet experiments to the manuscript.
>
>
> We are grateful for your valuable comments and suggestions which we have taken into full consideration in revising our manuscript. We look forward to hearing from you regarding our submission. If you have any further questions or comments, please do not hesitate to contact us.

---

### Review · Reviewer_yeKp · 2024-02-03

**Summary Of Contributions:**

This paper contributes a new way to construct a random sparse network, which aims to prevent the scenarios where some neurons monopolize all edges, or some edges are wasted due to dead neurons. The idea is to have a pre-processing step that ensures some minimum level of connectivity. The method works by first selecting the minimum set of neurons in each layer to be connected. Then, one adds connections so that each input and output node has at least one edge connected. Then, one adds additional edge until the desired number of edges are met. Experiments show that the proposed method can improve the performance of sparse models at extremely low rate.

**Audience:**

Yes

**Broader Impact Concerns:**

I do not think there should be a broader impact section for this paper.

**Claims And Evidence:**

Yes

**Requested Changes:**

- Given that the performance of the randomly pruned models are quite unstable (i.e., has a large variance), I strongly recommend the authors to perform experiments over multiple seeds and include the error bars. By doing so, one could even illustrate the advantages of this method better; MiCA should have better worst-case performance than RPI.
- I also recommend adding some diversity to the experiments, in terms of model architectures, datasets, and even tasks.

**Strengths And Weaknesses:**

### **Strengths**
- The main question that this paper aims to address, i.e., finding the additional conditions that help RPI work well, is interesting.
- The background of the research has been described comprehensively, placing the paper in the context well.
- The proposed method is quite simple, easy to implement, and does not require much computation.
- The proposed method indeed seems to work as intended, in the sense that it preserves the number of functional nodes well.


### **Weaknesses**
- The empirical performance of the method is, in my opinion, somewhat marginal. Most of the performance only takes place after a significant degradation of performance has been taken place, undermining the practical usefulness of the idea.
- The type of model architectures discussed in the paper is restricted; the paper uses VGG-16 and ResNet-20. As the proposed method is quite simple, I expect them to be applicable to other model architectures. But this has not been validated, which is a clear weakness.
- Similarly, the type of tasks and datasets considered is restricted and relatively small-scale. The paper tests on two datasets: CIFAR-10 and CIFAR-100. In fact, two datasets are too similar to each other (again, in my humble opinion), and does not fully provide the advantage of having two different datasets. I recommend adding more datasets, tasks (other than image classification), or even data modality.
- Also, the discussions and analyses could have been richer. For instance, I am quite curious whether the apparent gain comes from a better generalization, or a better training accuracy.

---

> ### Author Response · Authors · 2024-02-17
> **Official Comment by Authors**
>
> Thank you for your valuable feedback and constructive and helpful comments. We would like to respond to your comments as follows:
>
> > The empirical performance of the method is, in my opinion, somewhat marginal. Most of the performance only takes place after a significant degradation of performance has been taken place, undermining the practical usefulness of the idea.
>
> Thank you for raising this crucial point. It's true that asserting our method so that it achieves sufficient performance for practical applications in the high compression range might be challenging. However, given that our method specifically addresses scenarios where conventional PaI methods exhibit significant performance degradation, we urge our results to be viewed as indicative of our method's effectiveness. Furthermore, we would like to emphasize that the contributions of our paper not only enhance accuracy in the high compression range but also offers fresh insights into basic mechanism as to how PaI methods works. In the high compression range, the performance superiority of traditional PaI methods has shown variability based on factors such as PaI type and network structure, in clear contrast to our proposal. We believe this finding is novel within the area of PaI, as it elucidates the underlying reasons for this variation. We have revised the manuscript to provide a comprehensive overview of these contributions.
>
> > The type of model architectures discussed in the paper is restricted; the paper uses VGG-16 and ResNet-20. As the proposed method is quite simple, I expect them to be applicable to other model architectures. But this has not been validated, which is a clear weakness.
>
> > Similarly, the type of tasks and datasets considered is restricted and relatively small-scale. The paper tests on two datasets: CIFAR-10 and CIFAR-100. In fact, two datasets are too similar to each other (again, in my humble opinion), and does not fully provide the advantage of having two different datasets. I recommend adding more datasets, tasks (other than image classification), or even data modality.
>
> > I also recommend adding some diversity to the experiments, in terms of model architectures, datasets, and even tasks.
>
> Thank you for pointing this out. We have added the results of ImageNet experiments to the manuscript.
>
> > Also, the discussions and analyses could have been richer. For instance, I am quite curious whether the apparent gain comes from a better generalization, or a better training accuracy.
>
> Thank you for your interesting feedback. We have found that our method produces smaller training losses than other methods, and the trend is similar to the result of test accuracy in the manuscript. It reveals that learning in the high compression range becomes impossible in the first place if the layers are not sufficiently connected.
>
> > Given that the performance of the randomly pruned models are quite unstable (i.e., has a large variance), I strongly recommend the authors to perform experiments over multiple seeds and include the error bars. By doing so, one could even illustrate the advantages of this method better; MiCA should have better worst -case performance than RPI.
>
> Thank you for your very helpful suggestions. We have added error bars to the experimental results in the original manuscript. We believe the superiority of our method compared to RPI  is now more emphasized.
>
> We are grateful for your valuable comments and suggestions which we have taken into full consideration in revising our manuscript. We look forward to hearing from you regarding our submission. If you have any further questions or comments, please do not hesitate to contact us.

---

### Review · Reviewer_APXS · 2024-02-04

**Summary Of Contributions:**

This paper introduces a  pruning at initialization algorithm for high compression range: Minimum Connection Assurance (MiCA). Specifically, it preserves top-to-bottom information propagation among randomly pruned layers by building a random connection—minimum connection—using some of the pre-allocated edges.

**Audience:**

Yes

**Claims And Evidence:**

No

**Requested Changes:**

see the weakness.

**Strengths And Weaknesses:**

\+ The minimum connection is constructed by pre-determining and connecting the neurons that the subnetwork uses, and the subnetwork with the connection maintains the pre-defined sparsity distribution even when connecting its neurons randomly. Thus, all allocated edges can be functional even in a high compression range.

\+ In VGG-16 with CIFAR-10, MiCA improves the accuracy of random pruning by 27.0% at 104.7× compression ratio.

\- The novelty may be limited. It is mainly based on some observations and examples in Figure 2 and 3 to derive the inequalities. it does not have a well defined problem formulation, or formal proof. The rules it uses in the inequalities are straightforward. The technical contribution may be limited.

\- It is better to provide a detailed algorithm to formally describe the proposed method. For example, it selects  $|V_{out}^{l-1}|$  randomly from the range. If it selects the random output nodes for each layer, I am not sure how to make the final model follow certain predefined sparsity requirement. Although it lists some key steps, I am still not sure like what is the constraint for the model at beginning, what are the detailed steps from the beginning to the final stage to obtain the sparse model. It is better to have a clear definition about what is the constraint, what are the steps, what they want  to achieve and so on.

\- Although it claims to have a better accuracy for extremely sparse models, the accuracy of these extremely sparse models are still very low, and probably people may not use these models. It is better to discuss the application of the proposed method and the generated models.

\- The experiments are conducted on CIFAR-10 and CIFAR-100 for VGG and resnet models. I have some concerns about this part. Most pruning works have results on ImageNet. And VGG and resnet are very basic models. There are new models and architectures with better performance. The results on CIFAR with VGG or resnet does not seem to be strong.

\- I have some concerns about the results. In some cases, the improvements seem to be marginal, such as figure 6 (a) and (b) for resnet. The results of the proposed method are very close to the baselines. It seems not very effective on resnet.

\- It is better to discuss the advantages of the pruning at initialization. Maybe the training can be faster. But the authors need to have some results or discussions to explain they want to explore this area and investigate this problem. Currently I do not find a clear motivation.

---

> ### Author Response · Authors · 2024-02-17
> **Official Comment by Authors**
>
> Thank you for your insightful feedback and constructive and helpful comments. We would like to respond to your comments as follows:
>
> > The novelty may be limited. It is mainly based on some observations and examples in Figure 2 and 3 to derive the inequalities. it does not have a well defined problem formulation, or formal proof. The rules it uses in the inequalities are straightforward. The technical contribution may be limited.
>
> Thank you for your invaluable feedback. To elucidate our technical contribution, we have meticulously detailed the essence and specific algorithms our method seeks to accomplish. Our approach successfully ensures functional edges throughout all layers, even employing random connections, even in scenarios with remarkably high compression ratios. This achievement illuminates the essential prerequisites for networks to effectively learn in the high compression range. Specifically, it highlights the criticality of maintaining the layer-wise connections and maximizing the ratio of functional nodes. While the superiority of traditional PaI methods has demonstrated variability based on factors such as PaI type, network structure, and compression ratio, our finding unveils the underlying mechanisms governing these fluctuations. We believe that this insight contributes novel understanding to the area of PaI.
>
> > It is better to provide a detailed algorithm to formally describe the proposed method. For example, it selects $\left|V_{out}^{l-1}\right|$ randomly from the range. If it selects the random output nodes for each layer, I am not sure how to make the final model follow certain predefined sparsity requirement. Although it lists some key steps, I am still not sure like what is the constraint for the model at beginning, what are the detailed steps from the beginning to the final stage to obtain the sparse model. It is better to have a clear definition about what is the constraint, what are the steps, what they want to achieve and so on.
>
> Thank you for your informative remarks. We are fully convinced that the paper should have described a detailed algorithm that can resolve those uncertainties you have kindly pointed out. We have added a detailed algorithm for our method to the manuscript, which we believe successfully describe the proposed method in a more formal and clear manner.
>
> > Although it claims to have a better accuracy for extremely sparse models, the accuracy of these extremely sparse models are still very low, and probably people may not use these models. It is better to discuss the application of the proposed method and the generated models.
>
> Thank you for the important suggestion to improve our manuscript. We have added a discussion of applications to the conclusion section of the manuscript. The newly introduced discussion claims that our findings in the high compression range could contribute to areas of cost reduction of extremely large models such as foundation models and large language models as it will be necessary to compress models at a higher ratio beyond current practical limits to reduce the training and inference costs of large models. It also mentions the potential contribution in areas such as the fine-tuning of large models to downstream tasks by pruning.
>
> > The experiments are conducted on CIFAR-10 and CIFAR-100 for VGG and resnet models. I have some concerns about this part. Most pruning works have results on ImageNet. And VGG and resnet are very basic models. There are new models and architectures with better performance. The results on CIFAR with VGG or resnet does not seem to be strong.
>
> Thank you for pointing this out. We have added the results of ImageNet experiments to the manuscript.
>
> > I have some concerns about the results. In some cases, the improvements seem to be marginal, such as figure 6 (a) and (b) for resnet. The results of the proposed method are very close to the baselines. It seems not very effective on resnet.
>
> Thank you for your important remarks. Certainly, some of results of ResNet show only marginal improvements, but on some pre-defined sparsity distributions, our method clearly improves the compression ratio and accuracy trade-off. For example, for CIFAR-10, MiCA-ERK and MiCA-IGQ, and for CIFAR-100, MiCA-SNIP noticeably improve the trade-off for each RPI as illustrated in Figure 6 (a) and (b).
>
> Since the maximum number of characters has been reached, we will continue our reply in the next comment.

---

> > ### Author Response · Authors · 2024-02-17
> > **Official Comment by Authors**
> >
> > > It is better to discuss the advantages of the pruning at initialization. Maybe the training can be faster. But the authors need to have some results or discussions to explain they want to explore this area and investigate this problem. Currently I do not find a clear motivation.
> >
> > We are sorry that this part was not clear in the original manuscript. We have highlighted our motivation for studying pruning at initialization in the introduction of the manuscript.
> >
> > As mentioned in the first comment, we would like to emphasize that the contribution of this paper not only improves the accuracy in the high compression range but also provides new insights into the traditional primary mechanism by which the PaI method works.
> > Prior studies of PaI have only found that the advantage of each PaI depends on factors such as PaI type (RPI or RbPI), network structure, and compression ratio.
> > Our method outperforms conventional PaI independent of these factors, and further analysis suggests what is responsible for this advantage in the networks constructed by PaI.
> > We believe this is novel enough in the area of PaI.
> >
> > We are grateful for your valuable comments and suggestions which we have taken into full consideration in revising our manuscript. We look forward to hearing from you regarding our submission. If you have any further questions or comments, please do not hesitate to contact us.

---

### Decision · Action_Editor_8nLZ · 2024-03-26

**Recommendation:** Accept with minor revision

**Comment:**

The reviewers all agreed on submission meeting  `Claims And Evidence`  and  `Audience` criteria while raising concerns with current set of results and presentation.

In summary, motivation of the paper is well described and the question of finding conditions for random pruning at initialization to work well is an interesting question and would be of TMLR readers interest. While MiCA's effectiveness  are supported with convolutional networks on CIFAR-10/100 experiments,  reviewers also raised concerns about the generalizability of the paper's claim to architectures beyond convolutional networks and different dataset; moreover scalability which also related to usefulness / effectiveness of proposed method still remains as unanswered as well as concerns for reproducibility.

Here's some comment from reviewers in the reviews & recommendations for improvement of the paper:

`vomQ`: "the proposed subnetwork construction (two steps on page 5) is still ad hoc, which may not be principled enough to be adopted in different settings (models, datasets)", "work only test CIFAR-10 and CIFAR-100, which is limited.", "analysis will not be scalable once we switch to large-scale models of billions of parameters or hundreds of layers."

`APXS`: "runs the experiments for three times, ... using large amounts of randomness in the framework does not seem to be robust and the results may also be affected by randomness.", "results on ImageNet in Fig 7 show that the performance of the proposed method is very close to the baseline with marginal improvements"

`yeKp`: "gain over other methods is very small", "lacks a formal verification that the 'proposed algorithm' is an effective way.", "the advantages only takes place in not-so-practical sparsity level"

AE recommends considering following changes for improvement:

1) Address robustness over randomness with more repetition
2) Empirical demonstration of MiCA's effectiveness on compression <> accuracy tradeoff beyond convolutional networks
3) Demonstration of the method on general tasks: beyond CIFAR-10/100 (two are very similar type of tasks); ideally beyond just image classification task
4) Demonstration of scalability: e.g. effectiveness in ImageNet on both architecture with and without skip-connection (Note as reviewers pointed out current ImageNet experiments doesn't seem to convincingly support claimed improvements)
5) Better justification of why improvement of MiCA at high compression ratio is important despite still being quite low performance

As we understand these make take some effort to incorporate, for the changes that’s not able to achieve by camera ready, AE suggest the authors to clearly state these as limitations of the paper (e.g. in a limitations section).

**Audience:**

Reviewers pointed out that the motivation of the paper is well described and the question of finding conditions for random pruning at initialization to work well is an interesting question.

The topic the paper investigates is definitely of interest to a wide TMLR audience.

**Claims And Evidence:**

The authors propose a novel pruning at initialization (PaI) method, Minimum Connection Assurance (MiCA), for neural networks aimed at maintaining high performance at high compression ratios without the need for costly connection learning. Authors demonstrate  experimental results on CIFAR-10 and CIFAR-100 improving compression ratio and accuracy trade-offs.

One of the main claims from the authors is MiCA's effectiveness on 1) High Compression Range and 2) for networks with and without skip connection.  In this regard, authors show the effectiveness (in terms of compression ratio <> accuracy tradeoff) of MiCA in VGG (without skip-connection) and ResNets (with skip-connection) for CIFAR-10/CIFAR-100 datasets.

Reviewers raised concern on generalizability of the method as the empirical demonstration of effectiveness was limited to CIFAR-10/100 which is a relatively similar dataset and neural network architecture VGG and ResNets which are mostly convolutional networks.  Due to this limited demonstration, reviewers were concerned about the generalizability of the findings (whether the findings are too specific).

In the rebuttal authors have presented results on the ImageNet dataset, however, reviewers pointed out that the proposed method is very close to the baseline with marginal improvements. Therefore so far there's no demonstration that MiCA are effective on large scale datasets. Moreover validation on the methods beyond the convolutional architecture case still lacks after author rebuttal while the authors claim the generalizability. Moreover one reviewer pointed out that the proposed subnetwork construction is  somewhat "ad hoc" raising concern in general settings.

Relatedly one of the reviewer pointed out that benefit only shows in high-sparsity settings with significant degradation of performance and whether this improvement will ever be of practical interest.

Another issue raised by reviewers was the robustness of the algorithm; even after the rebuttal, few reviewers raised concern that randomness in the framework does not seem to be robust and the results may also be affected by randomness. Therefore concerns on reproducibility of the results remains while empirical validation in the submission is not satisfactory to assure reviewers that this is a robust algorithm.

---

> ### Author Response · Authors · 2024-04-24
> **Thank you for accepting our paper with minor revisions.**
>
> Dear Action Editor 8nLZ,
>
> Thank you for accepting our paper with minor revisions.
> In addition to the camera-ready version, we have released a version with the revised parts in red in the revision history.
> Also, the following is a summary of the revisions to the points you suggested.
> We hope these additional materials can help you in checking our final submission.
>
> > 1. Address robustness over randomness with more repetition
>
> We have increased the number of experiments from three to five.
> The revised results are basically similar to the original results.
> We have added our thoughts on the accuracy variation results in the limitations section.
> Also, we have added the error bar for the compression ratio to show that our method is robust to the corrected compression ratio.
>
> > 2. Empirical demonstration of MiCA's effectiveness on compression <> accuracy tradeoff beyond convolutional networks
>
> > 3. Demonstration of the method on general tasks: beyond CIFAR-10/100 (two are very similar type of tasks); ideally beyond just image classification task
>
> As an additional experiment, we have experimented with the node classification task, which is the major task for graph neural networks (GNNs).
> GNNs are based on MLP and do not use convolutional structures.
> We employ graph convolutional network (GCN) and graph isomorphism network (GIN) architecture.
> (Please note that while GCN seems to have a convolutional structure from the name, it does not have the convolutional structure contained in CNN.)
> Section 4.9 has discussed the results.
>
> > 4. Demonstration of scalability: e.g. effectiveness in ImageNet on both architecture with and without skip-connection (Note as reviewers pointed out current ImageNet experiments doesn't seem to convincingly support claimed improvements)
>
> > 5. Better justification of why improvement of MiCA at high compression ratio is important despite still being quite low performance
>
> We have mentioned these two requested revisions in the additional section, "Limitations and Implications of This Work."
>
> Again, thank you very much to the AE and the reviewers for their thoughtful comments and insightful feedback.
> We believe the paper has become substantially better by the suggested clarification and modifications.
> We hope this paper will be successfully published with this minor revision.
>
> Sincerely,
>
> Paper2038 authors